# Excellent Temperature-Control Based on Reversible Thermochromic Materials for Light-Driven Phase Change Materials System

**DOI:** 10.3390/molecules24081623

**Published:** 2019-04-24

**Authors:** Caixia Ren, Fangfang Liu, Malik Muhammad Umair, Xin Jin, Shufen Zhang, Bingtao Tang

**Affiliations:** 1State Key Laboratory of Fine Chemicals, Dalian University of Technology, Dalian 116024, China; 13127037258@163.com (C.R.); liufangfang@mail.dlut.edu.cn (F.L.); umairdut@mail.dlut.edu.cn (M.M.U.); zhangshf@dlut.edu.cn (S.Z.); 2Eco-chemical Engineering Cooperative Innovation Center of Shandong, Qingdao University of Science and Technology, Qingdao 266042, China; jinx1971@163.com

**Keywords:** photo-thermal conversion, phase change materials, thermochromic compound, temperature control

## Abstract

Light-driven phase change materials (PCMs) have received significant attention due to their capacity to convert visible light into thermal energy, storing it as latent heat. However, continuous photo-thermal conversion can cause the PCMs to reach high thermal equilibrium temperatures after phase transition. In our study, a novel light-driven phase change material system with temperature-control properties was constructed using a thermochromic compound. Thermochromic phase change materials (TC-PCMs) were prepared by introducing 2-anilino-6-dibutylamino-3-methylfluoran (ODB-2) and bisphenol A (BPA) into 1-hexadecanol (1-HD) in various proportions. Photo-thermal conversion performance was investigated with solar radiation (low power of 0.09 W/cm^2^) and a xenon lamp (at a high power of 0.14 W/cm^2^). The TC-PCMs showed a low equilibrium temperature due to variations in absorbance. Specifically, the temperature of TC-PCM_180_ (ODB-2, bisphenol A and 1-HD ratio 1:2:180) could stabilize at 54 °C approximately. TC-PCMs exhibited reversibility and repeatability after 20 irradiation and cooling cycles.

## 1. Introduction

As a fossil fuel substitute, solar energy is a renewable and clean alternative energy resource [1,2]. Solar energy utilization—including thermal utilization, photovoltaic conversion and photo-thermal conversion technology—has increasingly advanced [3,4,5,6,7]. However, thermal utilization of solar energy suffers from inferior direct heat usage factors due to inefficient utilization of visible light, which accounts for 44% of solar radiation [8,9]. To solve this problem, investigators have introduced light harvesting materials (such as dye, carbon materials and metal, etc.) into phase change material systems to obtain high photo-thermal conversion and energy storage efficiency [10,11,12,13,14]. On the other hand, continuous photo-thermal conversion can lead to high thermal equilibrium temperatures of phase change materials (PCMs) after phase transition [15,16,17]. This critical issue can have certain implications on the application of PCM systems based on photo-thermal conversion materials in some fields.

Typically, in a photo-thermal conversion system, light-harvesting materials absorb visible light, causing valence electrons to jump into an excited state and then deexcite to the ground state mainly by means of non-radiative transition [18,19]. Vast heat energy is emitted and then stored by PCMs [20]. In the process, as light radiates unremittingly, the temperature of PCM systems rises due to energy conversion until thermal equilibrium is attained. In a sense, the thermal equilibrium temperature depends on the balance between the photo-thermal conversion rate and heat conduction rate. Fortunately, thermochromic materials, which are typically composed of color former, developer and solvent, show superior performances in temperature control [21,22]. The tendency of thermochromic materials to become colorless and lose light absorption capacity with increases in temperature enables them to achieve quick thermal equilibrium at a low temperature. At present, thermochromism-induced methods of PCM-based systems have been studied widely to achieve energy storage or temperature control [23,24,25,26]. Peng Tang et al. designed thermochromism-induced temperature self-regulation and alternating photothermal nanohelix clusters, and applied these to synergistic chemo/phototherma therapy for tumours [26].

In the present study, 2-anilino-6-dibutylamino-3-methylfluoran (ODB-2) was selected as a color former. 1-hexadecanol (1-HD) and bisphenol A (BPA) served as solvent and developer, respectively. The thermochromic materials were black at room temperature, and when 1-hexadecanol melted into an amorphous state, the color faded. This process caused notable light absorption changes due to the color variation of ODB-2, and the thermal equilibrium under irradiation was achieved at a lower temperature, which was beneficial for increasing the photostability of the light-harvesting molecules under continuous irradiation. The mechanism is as follows: ODB-2, like many other similar chromophores, presents two different isomers: the leuco lactone form, which is colorless, and the open carboxylate state, which is colored. When 1-HD starts to crystalize as temperature decreases, the equilibrium between both isomers of ODB-2 is displaced by interactions with bisphenol A through hydrogen bonding, which favors ring-opening of the color former (Figure 1) [27,28,29,30].

In this work, thermochromic phase change materials (TC-PCMs) were obtained via combining ODB-2 and BPA with 1-HD in various ratios. Then, the photo-thermal conversion property was evaluated under different light sources: solar radiation (low power approximately 0.09 W/cm^2^) and xenon lamp (high power approximately 0.14 W/cm^2^). A similar trend was observed in both conditions: temperature can be controlled at a certain range. After 20 irradiation and cooling cycles, temperature control properties of TC-PCMs was found to be still achievable.

## 2. Results and Discussion

### 2.1. FT-IR Analysis

Figure 2 shows the Fourier Transform Infrared (FT-IR) spectra of 1-HD and TC-PCMs. The characteristic peak of COO^−^ at 1607 cm^−1^ attributed to the addition of thermochromic compound [31], appeared in all TC-PCMs’ curves. It confirmed the ring-opening structure of the lactone ring in ODB-2. Furthermore, the peaks at 3303 cm^−1^ (stretching vibration of O–H), 2956 cm^−1^ (C–H stretching vibration of –CH_3_), 2850 cm^−1^ (C–H stretching vibrations of –CH_2_) and 1062 cm^−1^ (C–O stretching vibration) are typical of 1-HD.

### 2.2. Thermal Property of TC-PCMs

Phase transition temperature and enthalpy play important roles in measuring the energy-storage performances of PCMs, which were characterized by differential scanning calorimetry (DSC). Figure 3 and Table 1 show the DSC curves and specific data on TC-PCMs, respectively. The 1-HD curve featured two exothermic peaks, which were caused by solid–solid transition at 42 °C (forming a rotation phase) [32] and liquid–solid transition at 48 °C, and an endothermic peak overlapped by two transitions (as mentioned above) [33]. The melting point (T_m_) of 1-HD was 48.76 °C, and the cooling points including liquid-solid transition (T_c_) and solid–solid transition (T_t_) were 48 °C and 41.87 °C, respectively.

The phase transition temperatures (T_c_ and T_m_) of TC-PCMs decreased subtly with increasing doping amounts (mass fraction of ODB-2 and BPA). Contrary to this, enthalpy values increase to varying degrees. This result can be attributed to the addition of a thermochromic compound, which promoted the formation of the rotator phase and crystalline structure of 1-HD [28]. Therefore, appropriate addition of ODB-2 and BPA had a negligible effect on the phase transition temperature and enhanced the enthalpy values of TC-PCMs.

### 2.3. X-ray Diffraction (XRD) Analysis

The crystallization property was characterized by X-ray Diffraction (XRD). As shown in curve a of Figure 4, typical diffraction peaks of 1-HD were observed at angles of 2θ = 20.6°, 21.4°, 21.8°, 22.1°, 24.2°, 24.7°, which corresponded to the TC-PCMs. In curves c–e, the peaks at 21.4° and 24.2° weakened—probably due to the doping of dyes. In curve b, the diffraction peaks were nearly unchanged, indicating that the doping dose of TC-PCM_180_ had no significant influence on the crystallinity of 1-HD.

### 2.4. Photo-Thermal Conversion Analysis

To study the temperature-control property of thermochromic materials, photo-thermal conversion was tested (Figure 5). Figure 5a,b shows the photo-thermal conversion curves of TC-PCMs and CB_180_. In Figure 5a, under sunlight irradiation (0.09 W/cm^2^), the thermal equilibrium temperature of TC-PCMs was evidently lower than that of CB_180_. This result was due to the structural transformation of the thermochromic compound when the temperature was higher than the phase transition temperature of 1-HD (Figure 5d) [22,33]. Furthermore, the thermal equilibrium temperature of TC-PCM_180_ stabilized at 56 °C in contrast with that of CB_180_, whose temperature exceeded 64 °C. Figure 5b shows the curves of TC-PCMs under the higher power of a simulative light source (IR removal, approximately 0.14 W/cm^2^). The same tendency was exhibited: TC-PCMs achieved lower thermal equilibrium temperatures. Therefore, as the lowest TC compound ratio (mass fraction of TC compound, 1.63%), TC-PCM_180_ can attain lower thermal equilibrium temperatures under high or low irradiating power. This result indicates that TC-PCM_180_ exhibits good temperature-control property for photo-thermal conversion.

In order to further investigate the temperature-control property of different PCMs with thermochromic compound, we selected 1-tetradecanol as the PCM, at an optimum ratio of 1:2:180 (ODB-2: BPA: 1-tetradecanol) under solar irradiation. The results are shown in Figure 5c. The curves of 1-tetradecanol and 1-HD reached different thermal temperatures of 51.5 °C and 43.4 °C respectively, which indicated that the system temperature could be controlled under irradiation.

### 2.5. Cycle Performance

The light reversibility and durability of TC-PCMs is a key factor to determine practical application in thermal storage systems. TC-PCMs underwent 20 irradiation and cooling cycles, and the photo-thermal conversion performance before and after cycling test is shown in Figure 6. The curves of TC-PCMs before and after 20 irradiation cycles were essentially consistent. The results indicate that TC-PCMs kept good photo-thermal conversion performance after 20 times cycles. And temperature can be controlled in a certain scope.

## 3. Materials and Methods

### 3.1. Materials

1-Hexadecanol (HD), 1-tertradecanol and bisphenol A (BPA, 2,2-*bis*(4-hydroxyphenyl)propane) were supplied by Tianjin Damao chemical reagent factory. 1-tertradecanol, 2-Anilino-6-dibutylamino-3-methylfluoran (ODB-2) was supplied by Shenzhen Duao science and technology Ltd. (Shenzhen, China). Carbon black purchased from Cabot Corporation (Boston, MA, USA) was modified with diazonium salt to disperse in the oil phase.

### 3.2. Preparation of TC-PCMs

Different ratios of TC-PCMs were obtained by mixing 1-HD, BPA and ODB-2 at weight ratios of 1:2:30, 1:2:60, 1:2:120 and 1:2:180 (TC-PCM_30_, TC-PCM_60_, TC- PCM_120_ and TC-PCM_180_, respectively). 1-HD was melted at 80 °C for 1 h via oil bath, then BPA and ODB-2 were dissolved in 1-HD with varying percentages in a 185 °C thermostatic oil bath for 2 min. CB_180_ was obtained by replacing ODB-2 with CB (carbon black) in TC-PCM_180_. 1-tertradecanol (replacing 1-hexadecanol at the ratio of 1:2:180) systems were obtained identically.

### 3.3. Characterization of TC-PCMs

FT-IR spectrophotometry (Nicolet 6700, Waltham, MA, USA) was used to evaluate the structure of TC-PCMs and 1-HD (using KBr pearls). The thermal properties of TC-PCMs and 1-HD were measured by DSC (TA Q20, New Castle, Delaware, USA), with a heating rate of 5°/min in an N_2_ atmosphere. The crystallinity of the samples was estimated through XRD (D/Max 2400, Rigaku, Tokyo, Japan) from 10° to 80°. The UV-visible absorption spectra of solid and liquid TC-PCMs were obtained using a Hitachi U-4100 (Tokyo, Japan) spectrophotometer and an Agilent HP 8453 (Santa Clara, CA, USA) spectrophotometer, respectively.

In the photo-thermal conversion property test: both solar radiation and a simulated light (CHF-XM35–500W xenon lamp with a parallel light source system, PerfectLight, Beijing, China) were used as photo sources. The xenon lamp produced light with continuous wavelength, ranging from 200 nm to 2000 nm, and its energy distribution in the visible light area is similar to those of the solar spectrum. In contrast, near-infrared light (700 nm to 1100 nm), which heightens energy distribution vastly, was removed by an IR-CUT filter. The light intensity of sunlight was approximately 0.09 W/cm^2^. 0.14 W/cm^2^ of simulated light (approaching the strongest amounts of solar radiation) was chosen to study photo-thermal conversion performances under high-intensity light. Meanwhile, the power was measured by an optical power meter (Perfect light PL-MW2000, Beijing, China). The temperature values were automatically recorded using a numerical control thermometer every 4 s.

## 4. Conclusions

In this research, a light-driven PCM system with temperature-control properties was constructed by preparing TC-PCMs. DSC analysis showed a slight shift in phase transition temperature of approximately 1 °C, and enthalpy increased to 247 J/kg (in TC-PCM_130_). The photo-thermal conversion test demonstrated that the ultimate temperature could be stabilized at specific values as the TC compound turned from black to a tinted shade during the phase change of 1-HD, which caused a drastic decrease in absorbance. With the lowest doping dose and minimum absorbance, TC-PCM_180_ reached 54 °C under a certain power range and exhibited the best temperature-control property. Furthermore, 1-tetradecanol systems were explored at an optimal ratio of 1:2:180, in which temperature can be controlled at the same range. In addition, the reversibility and durability of TC-PCMs remained unchanged after 20 irradiation and cooling cycles. Thus, the temperature-control performance of light-driven PCM systems was explored, and, based on thermochromic theory, excellent energy storage properties were achieved.

## Figures and Tables

**Figure 1 molecules-24-01623-f001:**
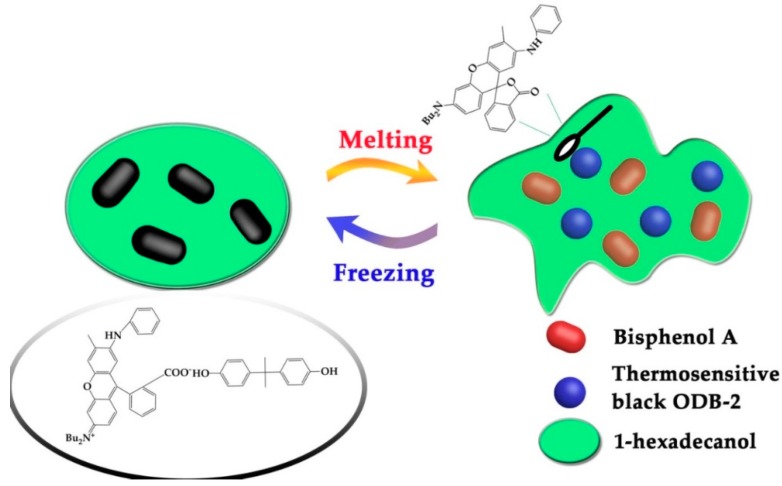
Diagram of the thermochromic process.

**Figure 2 molecules-24-01623-f002:**
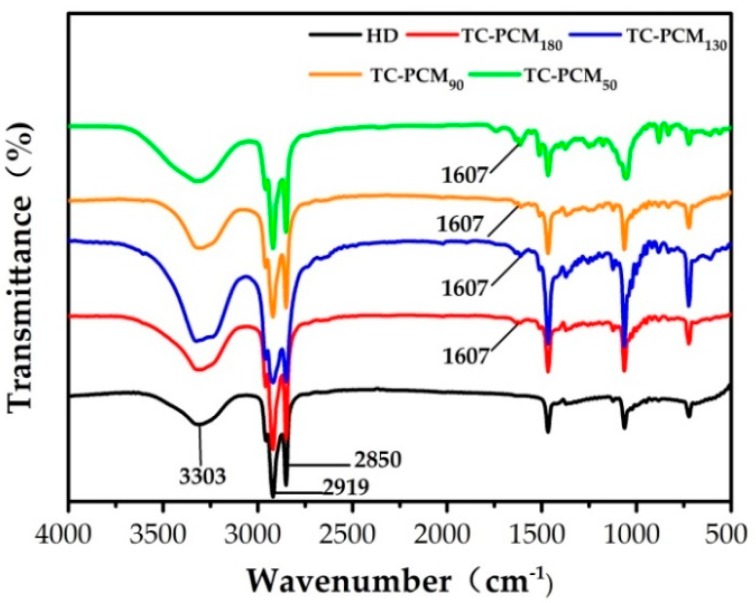
FT-IR spectra of 1-hexadedocanol (1-HD), thermochromic phase change material TC-PCM_180_, TC-PCM_130_, TC-PCM_90_ and TC-PCM_50._

**Figure 3 molecules-24-01623-f003:**
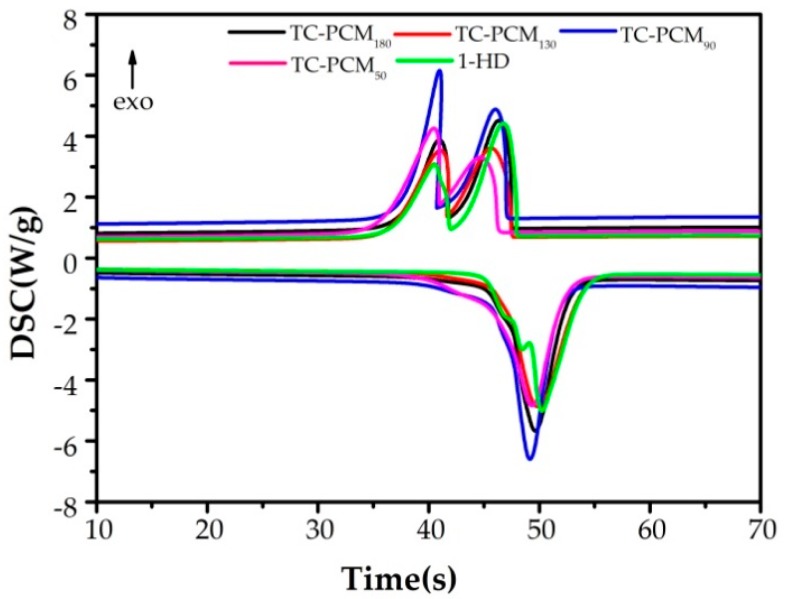
Curves of TC-PCM_180_, TC-PCM_130_, TC-PCM_90_, TC-PCM_50_, and pure 1-HD.

**Figure 4 molecules-24-01623-f004:**
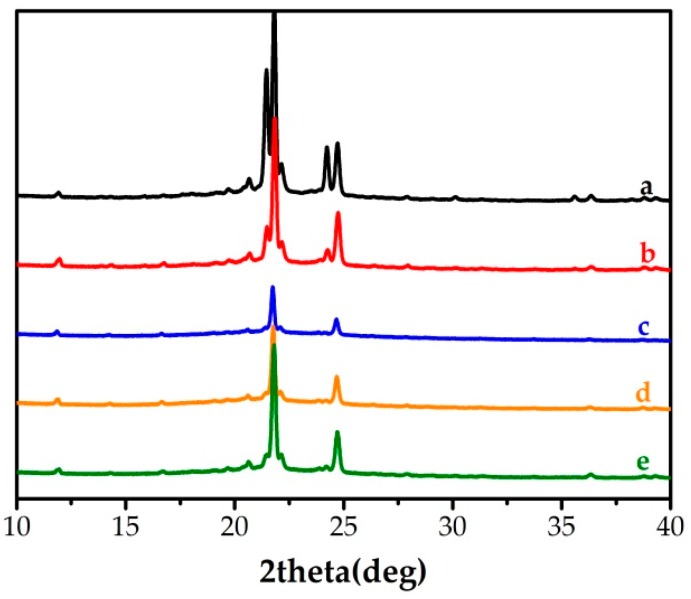
X-ray Diffraction (XRD) patterns of (a) 1-HD, (b) TC-PCM_180_, (c) TC-PCM_130_, (d) TC-PCM_90_, (e) TC-PCM_50_.

**Figure 5 molecules-24-01623-f005:**
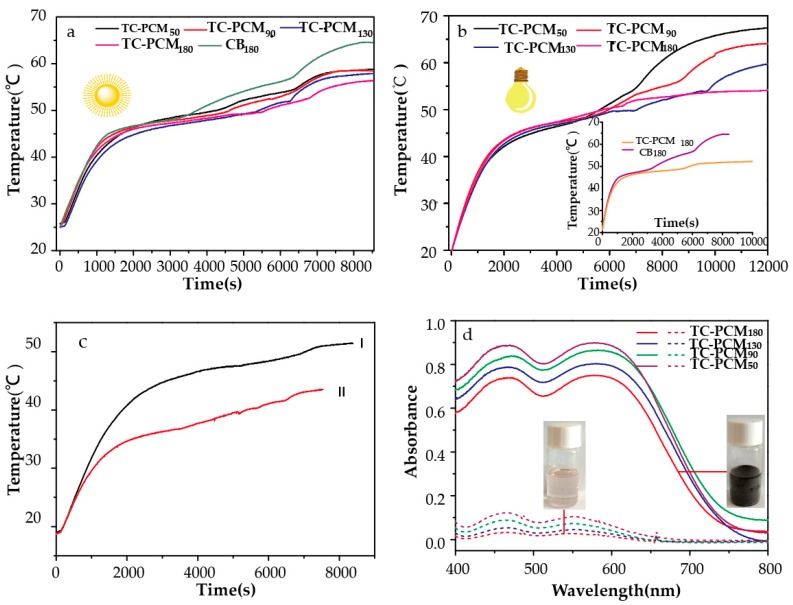
(**a**) Photo-thermal conversion curves under solar radiation (**b**) Photo-thermal conversion curves under simulative light (**c**) Photo-thermal conversion curves of 1-hexadecanol (І) and 1-tetradecanol (Ⅱ) as PCMs (**d**) visible light absorption of TC-PCMs under liquid and solid state.

**Figure 6 molecules-24-01623-f006:**
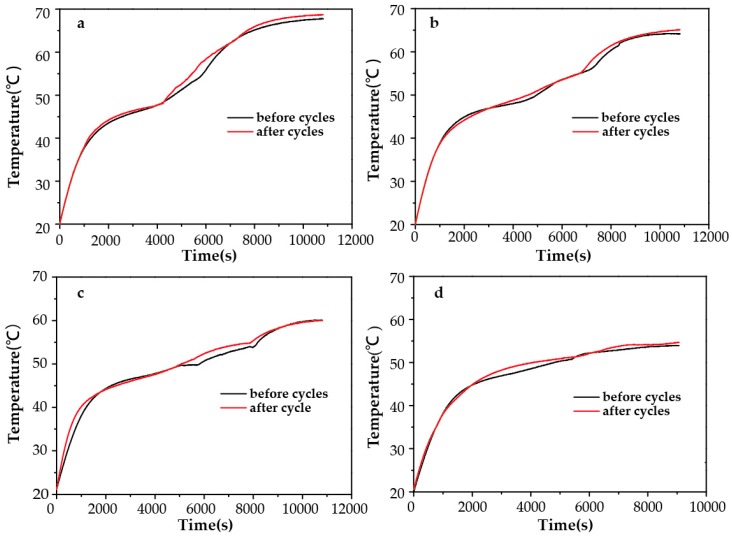
Photo-thermal conversion before and after 20 times irradiation and cooling cycles of (**a**) TC-PCM_50_ (**b**) TC-PCM_90_ (**c**) TC-PCM_130_ (**d**) TC-PCM_180_.

**Table 1 molecules-24-01623-t001:** Thermal date of TC-PCMs and pure 1-HD.

Sample	TC (% wt)	T_c_ (°C)	T_t_ (°C)	ΔH_c_ (J/g)	T_m_ (°C)	ΔH_m_ (J/g)
1-HD	0.00	48.00	41.87	229.5	48.76	233.8
TC-PCM_30_	1.89	46.23	41.12	242.9	46.04	244.5
TC-PCM_90_	1.08	47.01	40.81	244.9	46.95	240.3
TC-PCM_130_	0.75	47.47	41.85	248.9	47.03	247.2
TC-PCM_180_	0.55	47.45	41.81	236.3	47.13	238.7

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
