# Peer review of "Excellent Temperature-Control Based on Reversible Thermochromic Materials for Light-Driven Phase Change Materials System"

_molecules, 2019, doi:10.3390/molecules24081623_

Round 1

Reviewer 1 Report

The manuscript from Ren et al describes the development of a novel strategy to prepare light-driven phase change materials (PCM), where a thermochromic system that turns from colored to colorless upon PCM melting is used as a light-harvester. In this way, light absorption and, consequently, photothermal heating drastically decrease once surpassed the melting temperature of the PCM. As a result, lower temperatures are achieved under illumination with respect to conventional light-driven phase change materials. Although the idea is rather original, I cannot recommend publication of this article in Molecules in its present form based on the following reasons: 
1) Relevance of the work:a) Authors claim that their approach allows overcoming one of the main drawbacks of common photothermal conversion systems: when exposed to continuous illumination, their temperature also continuously increases even after melting of the PCM. In my opinion, this is not fully true. The temperature of these materials does not indefinitely raise under irradiation; instead, a thermal equilibrium is reached at a certain temperature that mainly depends on the rates of photothermal heating and of thermal diffusion towards the surrounding medium. Actually, this is observed in Figure 5a for the nonthermochromic CB150 sample, where a temperature plateau seems to be reached after 12000 s under solar radiation. To me, the main difference between their strategy and common photothermal conversion systems is that thermal equilibrium under irradiation is just achieved at a lower temperature above PCM melting point in their case. Therefore, the authors should have clearly discussed what the added advantage of this is. Moreover, I can think of other advantages for their approach which are not mentioned in the manuscript (e.g. increased photostability of the light-harvesting molecules under continuous irradiation) and could have been elaborated further by the authors. 
b) The use of photothermal effects to induce the thermochromic response of PCM-based systems has already described in the literature (e.g. Sci. Rep. 2016, 6, 36417; Biomaterials 2019, 188, 12-23). The authors fail to cite these works in their manuscript. Actually, they should have also referred to other photoinduced strategies to control the temperature transition of PCM materials (e.g. Nat. Commun. 2017, 8, 1446; Chem. Commun. 2018, 54, 10722). 

2) Contents of the work:a) I feel that the amount of work reported in the manuscript is not sufficient to warrant publication. Basically, the authors prepared four samples of thermochromic materials by dissolving different amounts of well-known color former and developer in a PCM, and then conducted very simple characterization measurements: IR and absorption spectra, XRD diffractograms, DSC thermograms, and temperature measurements under irradiation. In my opinion, the authors should have explored the application of their strategy to some other PCMs to demonstrate the generality of its scope. Furthermore, they should have demonstrated the reversibility and robustness of the materials prepared under consecutive cycles of photothermal heating and cooling in the dark. 
b) The discussion of the experimental results obtained also seems not fully adequate to me. For instance, nearly the same attention is devoted to discuss the IR data in Figure 2 (which in my opinion is unnecessary) as to comment on the measurements shown in Figure 5, which should constitute the central core of the manuscript. 
c) Several errors/inconsistencies are found in the way that the results are presented and discussed:- In some cases the authors incorrectly referred to the melting temperature of the PCM used as “molten temperature”. The same occurs when referring to the light absorption/absorbance of the samples analyzed, which they call “absorbency” through the manuscript. In a similar fashion: what do the authors refer when they write “thin solar energy and unavailability of visible light” (line 33), “non-radiation transitions” (line 43; it should non-radiative transitions) and “rotator structure of 1-HD”?- The inset shown in Figure 1 to illustrate the interaction between the color former and the color developer is wrong: why is there a positive charge on the hydroxyl group of bisphenol A? The authors should have properly checked previous literature when describing this interaction (e.g. Sci. Rep. 2017, 7, 16933).- The authors claim that no reaction takes place between the PCM (1-hexadecanol, 1-HD) and the color former and developer molecules based on the negligible changes observed by IR spectroscopy. However, would have the authors expected to detect any significant changes if reaction had taken place? To me, this would not have occurred, since the PCM is in a large excess in the thermochromic mixtures.- When discussing the changes observed in the thermal properties of the PCM upon addition of the color former and developer (Figure 3 and Table 1), the authors claim that “enthalpy values increased notably”. However, the maximum enhancement is only about 5%. About this point, what does the change in enthalpy not follow a linear trend with the concentration of the color former and developer? The authors should have also further discussed what the thermal transition at about 41º measured for 1-HD is.- The colors of the legends in Figure 5b do not fully correspond to the colors of the data sets plotted. In addition, legends are missing in Figure S1.- For Figure 5c to be fully meaningful, the absorption spectrum of the material before PCM melting should have also been shown.- When analyzing Figure 5b, I do not seem to observe very much differences for the final temperatures achieved for TC-PCM50 and CB180. Why is that?- For the TC-PCM180 sample, temperature stabilizes at around 54º under irradiation. The authors claim that this corresponds to 1-HD “molten temperature”. However, the melting temperature of this material is around 47-48º. How do the authors explain the difference?- In Figure 2 and 3, the abbreviation used for 1-hexadecanol is HC instead of 1-HD.- What is “simulated light” or “simulative light source”? I assume that the authors refer to a “solar simulator”, but they give no information about it in the “Materials and Methods” section. Even if they used another type of artificial light source, they should have clearly described it in this section.- The description of the materials prepared (and of the corresponding abbreviations) are given in the “Materials and Methods” section at the end of the manuscript. However, these abbreviations are used in previous sections without any definition, which makes it harder to read the manuscript. 
3) English language and style: Extensive editing of English language and style are required. 

Author Response

Thanks for you comments

Reviewer 2 Report

Tang et al. report on a series of thermochromic phase change materials based on the previously studied combination of 2-anilino-6-dibutylamino-3-methylfuran (ODB-2) and bisphenol-A (BPA). Specifically, through the use of 1-hexadecanol (1-HD) as a supporting matrix with an intermediate melting point, they explore strategies to develop temperature control by exploiting the phase change of the 1-HD and its effect on the thermochromic charge-transfer pair formed between ODB-2 and BPA in the solid state.

Unfortunately, given the poorly written and argued nature of their results (taking into account that the authors are unlikely to possess English as a first language) and a lack of care and scholarship in the interpretation of their data, I can only recommend that this manuscript is rejected for publication. This is so that the authors may take the time to improve their manuscript, prior to its resubmission in this or another more specialised journal. To aid the authors, I would recommend they consider the following points in particular:

·       As a general comment, the manuscript would benefit from proof-reading to improve its English. I understand that it is unlikely to be the authors first language but there are points where the poor writing did significantly affect my understanding of their work.

·       More specifically, I found the introduction to be weak. In my view it would benefit by helping the reader to understand more clearly how thermochromic phase change materials operate and how they differ from other thermochromic materials. What key roles the colour former, developer and solvent play? What is the current state of the art in temperature control in phase change materials? In addition, it was not at all clear to me what the novelty of their approach was – especially in that the ODB-2/BPA thermochromic charge-transfer complex has been explored previously. More detailed comment on this and a clearer assessment of the challenge their work addresses would be helpful.

·       The terminology used throughout the manuscript describing ‘simulated’ and ‘simulative light’ is not clear or detailed enough for me. What type of light exactly is it designed to simulate and what wavelengths is it operating at? Apart from its power density, what is the difference between this and the simulated solar light.

·       Figure 1 is incorrect. I do not agree with the proposed structure of their ODB-2/BPA complex – the OH group on the BPA should not carry a positive charge as drawn! That is an undergraduate mistake. Moreover, I would like to have seen the inactive version of the ODB-2 molecule in the figure as well since the lactone ring is clearly mentioned in the text.

·       I suggest the authors re-assess the interpretation of their IR data given that their proposed ODB-2/BPA complex is likely wrong.

·       It is never explained what CB180refers to, beyond appearing in Figure 5.

·       The significance of the ‘phase transition’ time (i.e. DT shown in Figure 5) is not clear to me and is given almost no mention in the text. I also don’t understand why it is shown for Fig 5a but not Fig 5b. The colours in the legend for Fig 5b do not match those in the graph. This, presumably, is a key performance metric for their new materials so why is it not described with any thought in the text?

·       There is no explanation for the temperature increases observed in almost all PCMs after the phase transition ‘plateau’ (i.e. the increase in T from ca. 50 degrees to 65 degrees in TC-PCM50 after ca. 4500 s irradiation with ‘simulative light’. Given that the point of this paper appears to be temperature control, I would like more attention to be paid to this trend in the data.

Author Response

Question 1: In my view it would benefit by helping the reader to understand more clearly how thermochromic phase change materials operate and how they differ from other thermochromic materials. it was not at all clear to me what the novelty of their approach was.

Thank you for your kindly suggestion.

Answer: Thermosensitive black ODB-2 as the color former shows good light absorption change. TC-PCMs are black, which has high absorbance in the room temperature (Figure 5d). When melting, TC-PCMs become colorless and the light absorbance declined vastly. Accordingly, temperature control property can be achieved.

Figure 5. (d) Absorbance of TC-PCMs under liquid and solid state

According to your suggestion, and in order to express the ideal more clearly, we added:

The thermochromic materials were black at room temperature, and when 1-hexadecanol melted into amorphous state, color faded. This process caused notable light absorption change due to the color variation of thermosensitive black ODB-2, and the thermal equilibrium under irradiation was achieved at a lower temperature above PCM melting point, which was beneficial to increased photostability of the light-harvesting molecules under continuous irradiation.

(Line 57, Page 2, Revised Manuscript)

Question 2: What key roles the colour former, developer and solvent play?

Thank you for your good question.

Answer: Lactone ring of color former opens ring, color exhibits. Developer serves as electron donor. Solvent controls the process via fusing and crystalizing.

According to your question, we detailed the process:

“1-HD starts to crystalize as temperature decreases, ODB-2 (electron donor) and bisphenol A (electron acceptor) are forced to transfer. During the migration process, bisphenol A contacts with ODB-2 and gains electron, then the molecular structure of ODB-2 was rearranged (lactone ring opens a ring) to form conjugate structure. Thus black color exhibits. During melting of 1-HD, bisphenol A attracts protons and parts with ODB-2then lactone ring closes and color fades (Figure 1) [28-31]. (Line 62, Page 2, Revised Manuscript)

Question 3: What is the current state of the art in temperature control in phase change materials?

Thank you for your good question.

Answer: At present, introducing thermochromic/ photo-induced materials in phase change materials is been studied widely. In particular, thermochromism-induced methods of PCM-based system can be applied to temperature regulation and biotherapy. Peng Tang et al designed thermochromism-induced temperature self-regulation and alternating photo-thermal system, then applying to synergistic tumor chemo/photothermal therapy. Grace G.D. Han et al combined photochromic molecules and organic phase change materials and realized long time energy at temperatures lower than the original crystallization point.

According to your suggestion, we supplemented this content in the revised manuscript.  At present, thermochromism-induced methods of PCM-based system has been investigated widely to achieve energy storage or temperature control [23-26]. Peng Tang et al designed thermochromism-induced temperature self-regulation and alternating photothermal nanohelix clusters, then applying to synergistic tumor chemo/phototherma therapy [26] .” (Line 50, Page 2, Revised Manuscript)

And we added references [23]-[26] in Section References

[23] Han, G.G.D.; Deru, J.H.; Cho, E.N.; Grossman, J.C. Optically-regulated thermal energy storage in diverse organic phase-change materials Chem. Comm. 2018, 54, 20722.

[24] Han, G.G.D.; Li, H.; Grossman, J.C. Optically-controlled long-term storage and release of thermal energy in phase-change materials. Nat. Commun. 2017, 8, 1446.

[25] Harrington, W.N.; Haji, M.R.; Galanzha, E.I.; Nedosekin, D.A. Photoswitchable non-fluorescent thermochromic dye-nanoparticle hybrid probes. Sci. Rep.2016, 6, 36417.

[26] Tang, P.; Liu, Y.; Liu, Y.; Meng, H.; Liu, Z.; Li, K.; Wu, D. Thermochromism-induced temperature self-regulation and alternating photothermal nanohelix clusters for synergistic tumor chemo/photothermal therapy. Biomaterials 2019, 188, 12-23.

Question 4: The terminology used throughout the manuscript describing ‘simulated’ and ‘simulative light’ is not clear or detailed enough for me. What type of light exactly is it designed to simulate and what wavelengths is it operating at? Apart from its power density, what is the difference between this and the simulated solar light.

Thank you for your kindly suggestion.

Answer:  “simulative light source” we used is CHF-XM35 -500W xenon lamp with a parallel light source system, whose emission spectra is 200-2000 nm and is similar to those of solar spectra in visible light region.

According to your good suggestion, we added the definition of simulative light source in “Materials and Method”. The specific modifications are as follows:

simulated light (CHF-XM35 -500W xenon lamp with a parallel light source system, PerfectLight) (Line 167, Page 7, Revised Manuscript)

The xenon lamp produced light with continuous wavelength, ranging from 200nm to 2000nm. And its energy distribution in visible light area is similar to those of solar spectrum. In contrast, near-infrared light (700 nm to 1100 nm) heightens vastly.” (Line 169, Page 7, Revised Manuscript)

Question 5: Figure 1 is incorrect. I do not agree with the proposed structure of their ODB-2/BPA complex – the OH group on the BPA should not carry a positive charge as drawn! That is an undergraduate mistake. Moreover, I would like to have seen the inactive version of the ODB-2 molecule in the figure as well since the lactone ring is clearly mentioned in the text.

Thank you for your kindly suggestion.

Answer: According to your suggestion, we modified Figure 1: the incorrect structure of positive charge on -OH was modified and structure of ODB-2 molecule was given.

Figure 1.Schematic diagram of thermochromic process

Question 6: I suggest the authors re-assess the interpretation of their IR data given that their proposed ODB-2/BPA complex is likely wrong.

Thank you for your kindly suggestion.

Answer: According to your comment, we examined the IR analysis and modified it.

“The characteristic peak of COO- at 40175px-1, attributed to the addition thermochromic compound [30], appeared in all TC-PCMscurves. It certified the ring-opening structure of lactone ring in ODB-2 when the color appeared.” (Line 80, Page 3, Revised Manuscript)

Question 7: It is never explained what CB180 refers to, beyond appearing in Figure 5.

Thank you for your kindly suggestion.

Answer: According to your comment, CB180 was given in materials part.

“CB180 was obtained by replacing ODB-2 with CB (carbon black) in TC-PCM180.” (Line 156, Page 6, Revised Manuscript)

Question 8: The significance of the ‘phase transition’ time (i.e. DT shown in Figure 5) is not clear to me and is given almost no mention in the text. I also don’t understand why it is shown for Fig 5a but not Fig 5b. The colours in the legend for Fig 5b do not match those in the graph.

Thank you for your kindly suggestion.

Answer: In our system, phase transition time depends on the enthalpy and photo-thermal conversion (light absorption performance). Long phase transition time means that the light absorption decreased thus photo-thermal heating rate decreased, which caused by the change of color. Low thermal equilibrium was reached based on the same reason.

We found the content repeated (low thermal equilibrium is more important in the manuscript). According to your comment, the content was deleted after our further study and consideration.

And I am sorry to make a mistake that colors in the legend for Fig 5b do not match those in the graph. And I have amended it.

Figure 5. (b) Photo-thermal conversion curves under simulative light

Question 9: There is no explanation for the temperature increases observed in almost all PCMs after the phase transition “plateau” (i.e. the increase in T from ca.50 degrees to 65 degrees in TC-PCMs after ca.4500s irradiation with “simulated light”. Given that the point of this paper appears to be temperature control, I would like more attention to be paid to this trend in the date.

Thank you for your kindly suggestion.

Answer: According to your suggestion, we added the reason of temperature increases in introduction and in the photo-thermal conversion analysis to emphasize the temperature control.

In the process, as light radiates unremittingly, the temperature of PCM systems rises via energy conversion until the thermal equilibrium will be achieved. In a sense, the thermal equilibrium temperature lies on the balance between photo-thermal conversion rate and heat conduction rate.” (Line 44, Page 2, Revised Manuscript)

“Same tendency exhibited: TC-PCMs achieved lower thermal equilibrium temperature. Therefore, As the lowest TC compound ratio (mass compound of TC compound, 1.63%), TC-PCM180 can attain lower thermal equilibrium temperature under high or low irradiating power.”

(Line 122, Page 4, Revised Manuscript) 

Reviewer 3 Report

This paper prepares new TC-PCM by using OBD-2, BPA, and 1-HD to study the photo-thermal conversion depending on different light sources. I suggest accepting the manuscript with major revisions. Some of the suggestions to improve the manuscript are:

-      Could you propose in the abstract and in the introduction some applications for these new formulated materials considering their phase change temperatures? Besides, deepen in the purpose to develop them.

-      Avoid realize as to do meaning.

-      The first time an acronym is given, it should be proper define. It means, in line 53 to 58, you should define the acronyms, although they were presented in the abstract.

-      Line 54: low temperature… which temperature, precisely?

-      Line 56: ODB-2 parts with BPA caused by 1-HD, better explain, as it is difficult to interpret this sentence.

-      Line 56: develop color, maybe exhibit is more precise?

-      Paragraph 52-57 should be included in Materials section

-      In my opinion, Materials and Methods section should be before than Results and discussion.

-      Figure 1 should be better described and explained. Indicate the acronyms for each used substance

-      The objectives should be also better defined.

-      When describing the used techniques to describe the samples, you should explain why the techniques are necessary.

-      The doping ratio is not well defined.

-      Lack of units in FT-IR analysis description.

-      Is FT-IR performed by means ATR? Or KBr pearls?

-      Figure 2 is incorrect. X axis should be presented in the opposite scale (from 4000 to 500 cm-1). Y axis is not absorbance, as it is resented. Y axis is transmittance (%).

-      Line 91: influenced in the crystalline…, in which sense? It became more rapid?

-      Line 92: the 0.75 % addition, addition of what? Where is this percentage exposed and explained?

-      Line 94-95: are greyly marked, why?

-      Line 95: better explain the sentence.

-      Figure 3 and Table 1 is double information for the same experiment. You can add the T in X axis.

-      Line 114-115: the sentence has no sense, due to the lack of some part of the sentence.

-      Figure 5 b: why the CB180 graph is not included in the general graph?

-      How many replicates have you performed for each material?

-      Section 3.3: This section should be presented in the same order as the characterization was presented in the previous section.

-      Line 144: you expose you have performed TGA. Where are the results?

-      Line 146: the heating ramp was from 0ºC to 800 ºC or room temperature to 800 ºC? The units for temperature are “ºC” or “ºF” or “K”, but never “º”.

-      Line 148-149: where are the UV results presented?

-      In conclusions section you should also present the FT-IR, DRX, TGA, DSC… results; or at least, mention some other results. Besides, could you please specify which is the best formulation, or the best combination of your TC-PCM?

Author Response

Question 1: Could you propose in the abstract and in the introduction some applications for these new formulated materials considering their phase change temperatures? Besides, deepen in the purpose to develop them.

Thank you for your kindly suggestion.

Answer: According to your helpful comments, we supplement the applications in Introduction:

Furthermore, the system demonstrates good foreground in protecting hot-side of thermoelectric system, based on its property of temperature control and protecting organic phase change materials.(Line 73, Page 2, Revised Manuscript)

“This process caused notable light absorption change due to the color variation of thermosensitive black ODB-2, and the thermal equilibrium under irradiation was achieved at a lower temperature above PCM melting point, which was beneficial to increased photostability of the light-harvesting molecules under continuous irradiation.” (Line 58, Page 2, Revised Manuscript)

Question 2: The first time an acronym is given, it should be proper define. It means, in line 53 to 58, you should define the acronyms, although they were presented in the abstract.

Thank you for your kindly suggestion.

Answer: According to your excellent question, we define the acronyms.

“2-anilino-6-dibutylamino-3-methylfluoran (thermosensitive black ODB-2) was selected as color former [25]. 1-hexadecanol (1-HD) and bisphenol A (BPA) served as solvent and developer, respectively.”  (Line 55, Page 2, Revised Manuscript)

Question 3: Line 54: low temperature… which temperature, precisely?

Thank you for your good question.

Answer: “low temperature” in the manuscript means temperature below the melting point. However, the expression was not clear. According to your question, we changed the expression. 1-HD starts to crystalize as temperature decreases” (Line 62, Page 2, Revised Manuscript)

Question4: ODB-2 parts with BPA caused by 1-HD, better explain, as it is difficult to interpret this sentence.

Thank you for your kindly reminder.

Answer: When 1-HD melted into amorphous state, bisphenol A could attract protons melted and ODB-2 was closed in the ring, thus the color disappeared. According to your question, we supplemented some details.

1-HD starts to crystalize as temperature decreases, ODB-2 (electron donation) and bisphenol A (electron gain) are forced to transfer. During the migration process, bisphenol A contacts with ODB-2 and gains electron, then the molecular structure of ODB-2 was rearranged (lactone ring opens a ring) to form conjugate structure. Thus black color exhibits. During melting of 1-HD, bisphenol A attracts protons and parts with ODB-2then lactone ring closes and color fades (Figure 1) [28-31].

 (Line 62, Page 2, Revised Manuscript)

Question 5: develop color, maybe exhibit is more precise.

Thank you for your kindly suggestion.

Answer: To express more clearly, according to your proposal, we chose “exhibit”.

Thus black color exhibits.(Line 65, Page 2, Revised Manuscript)

Question 6: Paragraph 52-57 should be included in Materials section.

Thank you for your kindly suggestion.

Answer: In a view of understanding easily and other references, we decided put the reaction mechanism in original site.

Question 7: In my opinion, Materials and Methods section should be before than Results and discussion.

Thank you for your kindly suggestion.

Answer: We intended to put the Materials and Methods section before Results and discussion, however, the requirement of molecules is that Materials and Methods is prior to Results and discussion.

Question 8:  Figure 1 should be better described and explained. Indicate the acronyms for each used substance.

Thank you for your kindly suggestion.

Answer: According to your suggestion, we described and explained newly.

1-HD starts to crystalize as temperature decreases, ODB-2 (electron donation) and bisphenol A (electron gain) are forced to transfer. During the migration process, bisphenol A contacts with ODB-2 and gains electron, then the molecular structure of ODB-2 was rearranged (lactone ring opens a ring) to form conjugate structure. Thus black color exhibits. During melting of 1-HD, bisphenol A attracts protons and parts with ODB-2then lactone ring closes and color fades (Figure 1) [28-31].

(Line 62, Page 2, Revised Manuscript)

Question 9:  The objectives should be also better defined.

Thank you for your kindly suggestion.

Answer: Our objective is to express the temperature reaches low thermal equilibrium temperature, thus temperature control property can be achieved. 

According to your important comments, we added and emphasized our objective.

Temperature can be controlled at a certain range. When 20 irradiation and cooling cycles were performed, temperature control still can be achieved.”  (Line 71, Page 2, Revised Manuscript)

Question 10: When describing the used techniques to describe the samples, you should explain why the techniques are necessary.

Thank you for your kindly reminder.

Answer: According to your important comments, we added reasons of using the techniques:

FT-IR: It certified the ring-opening structure of lactone ring in ODB-2 when the color appeared.” (Line 80, Page 2, Revised Manuscript)

DSC: “Phase transition temperature and enthalpy play important roles in measuring energy-storage performance, which was characterized by differential scanning calorimetry (DSC).” (Line 89, Page 3, Revised Manuscript)

XRD: “The crystallization property was certified by XRD.” (Line 106, Page 4, Revised Manuscript)

Photo-thermal conversion: “To study the temperature-control property of thermochromic materials, photo-thermal was tested.” (Line 114, Page 5, Revised Manuscript)

Question 11: The doping ratio is not well defined.

Thank you for your kindly reminder.

Answer: According to your important comments, we defined the doping ratio as mass compound of TC compound” (Line 123, Page 5, Revised Manuscript)

Question 12:  Lack of units in FT-IR analysis description.

Thank you for your kindly reminder.

Answer: According to your important comments, we added units in FT-IR analysis

“The bands at 3303 cm-1 (stretching vibration of O–H), 2956 cm-1 (C–H stretching vibration of–CH3), 2850 cm-1 (C–H stretching vibrations of –CH2) and 1062 cm-1 (C–O stretching vibration) are typical in 1-HD (Figure 2 d).” (Line 80, Page 3, Revised Manuscript)

Question 13: Is FT-IR performed by means ATR? Or KBr pearls?

Thank you for your good question.

Answer: The FT-IR spectrum of TC-PCMs and 1-HD were tested using KBr pearl.

According to your important question, we added KBr pearls in FT-IR.

“FT-IR spectrophotometry (Nicolet 6700, America) was used to evaluate the structure of TC-PCMs and 1-HD (using KBr pearls).” (Line 160, Page 6, Revised Manuscript)

Question 14: Figure 2 is incorrect. X axis should be presented in the opposite scale (from 4000 to 500 cm-1). Y axis is not absorbance, as it is resented. Y axis is transmittance (%).

Thank you for your kindly suggestion.

Answer: According to your important comments, we amended the Figure.

X axis was presented in the opposite scale (from 4000 to 500 cm-1). Y axis was transmittance (%).

Figure 2.The FTIR spectra of HD, TC-PCM180, TC-PCM130, TC-PCM90, TC-PCM50

Question15:   Line 91: influenced in the crystalline…, in which sense? It became more rapid?

Thank you for your good question.

Answer: From Table 1, we can see that enthalpy values increased when doping thermochromic compound. So it is in favor of crystalline.

According to your important comments, we supplemented thatwhich promoted the formation of rotator and crystalline structure of 1-HD [29] (Line 97, Page 3, Revised Manuscript)

Table 1. Thermal date of TC-PCMs and pure 1-HD.

Sample          TC (wt %)   Tc()    Tt()   ΔH c (J/g)   Tm(℃)   ΔH m(J/g)

HD             0.00        48.00     41.87     229.5      48.76       233.8

TC-PCM30           1.89        46.23     41.12     242.9      46.04       244.5

TC-PCM90           1.08        47.01     40.81     244.9      46.95       240.3

TC-PCM130           0.75        47.47     41.85     248.9      47.03       247.2

TC-PCM180          0.55              47.45     41.81     236.3      47.13       238.7

Question16: Line 92: the 0.75 % addition, addition of what? Where is this percentage exposed and explained?

Thank you for your good question.

Answer: the 0.75% addition is the mass fraction of ODB-2 and BPA.

According to your and other reviews’ comment, when modifying the manuscript, the contents were deleted.

Question17:   Line 94-95: are greyly marked, why?

Thank you for your good question.

Answer: I am sorry that it is a mistake and we have corrected.

Question18:   Line 95: better explain the sentence.

Thank you for your kindly reminder.

Answer: From Table 1, we can see that the Tc and Tm of TC-PCMs have little shift compared with 1-HD, and the enthalpy values increase by 5% at most. It may be caused by the addition of BPA and ODB-2.

According to your suggestion, we added some contents based on the manuscript.

“The phase transition temperatures (Tc and Tm) of TC-PCMs decreased subtly with increasing doping amounts (mass fraction of ODB-2 and BPA). And enthalpy values were in varying degrees of increase. This result can be attributed to the addition of thermochromic compoundwhich promoted the formation of rotator and crystalline structure of 1-HD [29].” (Line 96, Page 3, Manuscript)

Table 1. Thermal date of TC-PCMs and pure 1-HD.

Sample          TC (wt %)   Tc()    Tt()   ΔH c (J/g)   Tm(℃)   ΔH m(J/g)

HD             0.00        48.00     41.87     229.5      48.76       233.8

TC-PCM30           1.89        46.23     41.12     242.9      46.04       244.5

TC-PCM90           1.08        47.01     40.81     244.9      46.95       240.3

TC-PCM130           0.75        47.47     41.85     248.9      47.03       247.2

TC-PCM180          0.55              47.45     41.81     236.3      47.13       238.7

Question19: Figure 3 and Table 1 is double information for the same experiment. You can add the T in X axis.

Thank you for your kindly suggestion.

Answer: Considering that the table 1 shows detailed and specific information, including enthalpy values and melting temperature and crystalline temperature, we think that Figure 3 and Table 1 should remain.

Question 20:  Line 114-115: the sentence has no sense, due to the lack of some part of the sentence.

Thank you for your kindly reminder.

Answer: According to your and other reviews’ important comments, we deleted the expression:

(“This result can be attributed to the fading of TC-PCMs during the phase change stage of 1-HD with declining absorbency (S 1)”) (Line 113, Page 4, Manuscript)

Question 21: Figure 5 b: why the CB180 graph is not included in the general graph?

Thank you for your excellent question.

Answer: In the Figure 5b, we focused on difference of the temperature control property between TC-PCMs, so we put the comparison curves between CB180 and TC-PCM 180 in a small graph.

Question 22: How many replicates have you performed for each material?

Thank you for your good question.

Answer: When we test the photo-thermal conversion, we repeat it two or more times under different light power.  

According to your and other reviews’ important comments, we added the cycles test. And the results are as follows.

Figure 6. Photo-thermal conversion before and after 20 times irradiation and cooling cycles of a) TC-PCM50 b)TC-PCM90 c)TC-PCM130 d)TC-PCM180

“The light reversibility and durability of TC-PCMs impacted their application persistence. The TC-PCMs underwent 20 irradiation and cooling cycles, and the photo-thermal results were shown in Figure 6. The curves of TC-PCMs before and after 20 irradiation cycles were consistent basically. The results indicate that TC-PCMs kept good photo-thermal conversion performance after 20 times cycles. And temperature can be controlled in a certain scope.” (Line 138, Page 6, Revised Manuscript)

Question23: Section 3.3: This section should be presented in the same order as the characterization was presented in the previous section.

Thank you for your kindly reminder.

Answer: According to your important comments, we changed the order as you say.

Question24:  Line 144: you expose you have performed TGA. Where are the results?

Thank you for your excellent question.

Answer: Sorry to make the mistake because we reconsidered the importance of TGA analysis, and deleted it in manuscript, however, forgetting to remove it in the materials and methods part.

Question 25: Line 146: the heating ramp was from 0ºC to 800 ºC or room temperature to 800 ºC? The units for temperature are “ºC” or “ºF” or “K”, but never “º”.

Thank you for your kindly reminder.

Answer: Thanks for your comment on TGA.

Question 26: Line 148-149: where are the UV results presented?

Thank you for your good question.

Answer: The UV results were used to certify the spectral shifts in absorption. The UV results were shown in Figure 5d to certify the spectral shifts in absorption.

Figure 5. (d) Absorbance of TC-PCMs under liquid and solid state

Question 27: In conclusions section you should also present the FT-IR, DRX, TGA, DSC… results; or at least, mention some other results. Besides, could you please specify which is the best formulation, or the best combination of your TC-PCM?

Thank you for your kindly suggestion.

Answer: According to your important comments, we added some analysis and best combination of your TC-PCM.

“With the lowest doping dose and minimum absorbance, TC-PCM180 reached 54 °C under a certain power range and exhibited best temperature- control property.  Furthermore, 1-dodecanol and 1-tetradecanol systems were explored at the ratio of 1:2:180, in which temperature can be controlled at the same range. The reversibility and durability of TC-PCMs remained after 20 irradiation and cooling cycles.”

(Line 183, Page 6, Revised Manuscript)

Round 2

Reviewer 1 Report

Although the authors have addressed most of the points raised in my previous report, the manuscript still contains several flaws and inconsistencies that must be addressed before any positive assessment on their submission can be made: 
1) Mechanism of the thermochromic behavior of ODB-2/BPA mixtures in 1-HD: as required in my previous report, the authors corrected the error in Figure 1 concerning the interaction between ODB-2 and BPA that results in ODB-2 ring opening and coloration in solid state 1-HD. However, they have now included an explanation in the text which is completely wrong. According to the authors, an electron transfer takes place between ODB-2 and BPA:“The electron transfer mechanism is as follows: 1-HD starts to crystalize as temperature decreases, ODB-2 (electron donation) and bisphenol A (electron gain) are forced to transfer. During the migration process, bisphenol A contacts with ODB-2 and gains electron, then the molecular structure of ODB-2 was rearranged (lactone ring opens a ring) to form conjugate structure. Thus black color exhibits. During melting of 1-HD, bisphenol A attracts protons and parts with ODB-2, then lactone ring closes and color fades (Figure 1)”. (lines 62-67)This is not true. There is no electron transfer between ODB-2 and BPA. It is much simpler than that. ODB-2, as many other similar chromophores, presents two different isomers: the leuco lactone form, which is colorless, and the open carboxylate state, which is colored. The equilibrium between both isomers can be displaced by interactions with the surrounding media (e.g. protonation, hydrogen-bond formation). This is the case of BPA: a complex is formed between this species and the open form of ODB-2 by hydrogen bonding, which favors ring-opening of the color former. As I already recommended in my previous report, the authors should have properly checked previous literature when describing this interaction (e.g. Sci. Rep. 2017, 7, 16933). 
2) The last sentence of the introduction reads: “Furthermore, the system demonstrates good foreground in protecting hot-side of thermoelectric system, based on its property of temperature control and protecting organic phase change materials.” Unfortunately, the authors show no data in their manuscript to support this claim. As such, it should be removed. 
3) Line 99: “…, which promoted the formation of rotator and crystalline structure of 1-HD.” What does “rotator” refers to in this sentence? It is not clear at all to me. 
4) As suggested, the authors included additional measurements on the revised version of their manuscript where they investigated the light-driven thermochromic process in other PCMs (1-dodecanol and 1-tetradecanol), which have quite lower melting temperatures than 1-HD (about 24 and 38ºC, respectively). This means that the colored-colorless transitions of these materials should take place at those lower temperatures. Is it so? However, it is rather surprising that the maximum temperatures that are achieved under continuous irradiation are very similar for 1-HD, 1-dodecanol and 1-tetradecanol. How can the authors explain that? Actually, this result undermines the main claim of the article: that the use of the color-changing thermochromic system allows control on the final temperature reached under illumination, which does not seem the case now. For this reason, clear and thorough discussion of the results obtained in these new experiments is needed. 
5) Although the authors claim that native English speakers revised their manuscript, plenty of spelling and grammar errors can still be found (e.g. line 51: …  of PCM-based system  been investigated …; line 81: ... to the  thermochromic compound; line 82: Furthermore,  bands …; and many more). Therefore, extensive editing of English language and style are required.

Author Response

Question 1: Mechanism of the thermochromic behavior of ODB-2/BPA mixtures in 1-HD: as required in my previous report, the authors corrected the error in Figure 1 concerning the interaction between ODB-2 and BPA that results in ODB-2 ring opening and coloration in solid state 1-HD. However, they have now included an explanation the text which is completely wrong. According to the authors, an electron transfer take place between ODB-2 and BPA: “The electron transfer mechanism is as follows: “1-HD starts to crystalize as temperature decreases, ODB-2 (electron donation) and bisphenol A (electron gain) are forced to transfer. During the migration process, bisphenol A contacts with ODB-2 and gains electron, then the molecular structure of ODB-2 was rearranged (lactone ring opens a ring) to form conjugate structure. Thus black color exhibits. During melting of 1-HD, bisphenol A attracts protons and parts with ODB-2then lactone ring closes and color fades (Figure 1).” (lines 62-66) This is not true. It is much simpler than that. ODB-2, as many other similar chromophores, present two different isomers: the leuco lactone form, which is colorless, and the open carboxylate state, which is colored. The equilibrium between both isomers can be displaced by interactions with the surrounding media (e.g. protonation, hydrogen-bond formation). This is the case of BPA: a complex is formed between this species and open form of ODB-2 by hydrogen bonding, which favors ring-opening of the color former.

Thank you for your suggestion.

Answer: According to your comments, we consulted the relevant literatures and modified the thermochromic mechanism of as-prepared PCMs. Following modification was made: The mechanism is as follows: ODB-2, as many other similar chromophores, present two different isomers: the leuco lactone form, which is colorless, and the open carboxylate state, which is colored. When 1-HD starts to crystalize as temperature decreases, the equilibrium between both isomers of ODB-2 is displaced by interactions with bisphenol A through hydrogen bonding, which favors ring-opening of the color former (Figure 1) [27-30].” (Line 62, Page 2, Revised manuscript)

Question 2: The last sentence of the introduction reads:Furthermore, the system demonstrates good foreground in protecting hot-side of thermoelectric system, based on its property of temperature control and protecting organic phase change materials.”

Unfortunately, the authors show no data in their manuscript to support this claim. As such, it should be removed.

Thank you for your kindly suggestion.

Answer: According to your suggestion, we deleted it.

Question 3: “Which promoted the formation of rotator and crystalline structure of 1-HD”? What does “rotator” refers to in this sentence? It is not clear at all to me. 

Thank you for your excellent question.

Answer: For the n-alkanols with between 12 and 20 carbon atoms, two ordered monoclinic phases at room temperature, named γ and β. Before melting, these ordered phases transform to rotator phases where the molecules rotate around their long axis (solid - solid transition). (Chem. Mater. 2002, 14, 508-517, Appl. Energ. 2018, 212, 455-464)

Following your suggestion we have clarified the sentence by adding following information: “The 1-HD curve featured two exothermic peaks, which were caused by solid - solid transition at 42 °C (forming a rotate phase) [33]and liquid-solid transition at 48 °C, …” (Line 88, Page 3, Revised manuscript)

Question 4: As suggested, the author included additional measurements on the revised version of their manuscript where they investigated the light-driven thermochromic process in other PCMs (1-dodecanol and 1-tetradecanol), which have quite lower melting temperature than 1-HD (about 24 and 38 ℃,respectively). This means that the colored-colorless transitions of these materials should take place at those lower temperatures. Is it so? However, it is rather surprising that the maximum temperatures that are achieved under continuous irradiation are very similar for 1-HD, 1-dodecanol and 1-tetradecanol. How can the authors explain that? Actually, this result undermines the main claim of the article: that the use of the color-changing thermochromic system allows control on the final temperature reached under illumination, which does not seem the case now. For this reason, clear and thorough discussion of the results obtained in these new experiments is needed.

Thank you for your good question.

Answer: In the article, we want to express that thermochromic materials can reach thermal equilibrium at a low temperature to achieve long-term temperature control. When melting, the materials become colorless, resulting in decrease in absorbance, and then light absorption rate decreases. The temperature can stabilize at the melting point, achieving temperature control in a certain amount of time.  

The enthalpy values of 1-dedocanol, 1-tetradecanol and 1-HD are 186J/g, 210J/g and 233.8J/g, respectively (Table S 1). In view of low enthalpy values of 1-dodecanol and strong UV-light of the simulative light source, where the thermochromic materials showed strong absorption, we supplemented the photo-thermal conversion test of 1- HD and 1-tetradecanol under radiation of sunlight. We can see that the temperature of 1-HD and 1-tetradecanol stabilized at 51, 43, respectively.

In order to further investigate the temperature-control property of different PCMs with thermochromic materials, we selected 1-tetradecanol as the PCMs at an optimum ratio of 1:2:180 (ODB-2: BPA: 1-tetradecanol) under the solar irradiation. The results are shown in Figure 5c. The curves of 1-tetradecanol and 1-HD reached thermal temperature of 51 °C and 43 °C respectively, which indicated temperature can be control under irradiation. (Line 126, Page 5, Revised manuscript)

Table S 1 Enthalpy of fusion of 1-HD, 1-tetradecanol and 1-dedocanol

      ΔH m(J/g)

1-HD

233.8J/g

1-tetradecanol

210.1J/g

1-dedocanol

186J/g

Figure 5c Photo-thermal conversion of, 1-tetradecanol () and 1-hexadecanol () as PCMs under solar radiation

Question 5: Although the authors claim that native English speakers revised their manuscript, plenty of spelling and grammar errors can still be found (e.g. line 51: of PCM-based system has been investigated; line 81: to the thermochromic compound; line 82: Furthermore, bands; and any more). Therefore, extensive editing of English language and style are required.

Thank you for your suggestion.

Answer: A proof reading was conducted to improve the language and style. And we revised our manuscript. The modification of language is marked by “  ”

There are some substitutions of words, such as changing “Certified” (Line 82, manuscript) to “confirmed” (Line 79, revised manuscript), “Uninterrupted” (Line 15, manuscript) to “continuous” (Line 15, revised manuscript)  “Superiority” (Line 48, manuscript)  to “superiority performance” (Line 48, revised manuscript) etc.

There are some adjustments of sentences, for example, we changed “Therein thermal utilization exhibits inefficient direct heat usage factor owing to low thermal efficiency and inefficient utilization of visible light, which accounts for 44% of solar radiation.” (Line 32, manuscript) to “However, the thermal utilization suffers from inferior direct heat usage factor due to inefficient utilization of visible light, which accounts for 44% of solar radiation [8, 9].” (Line 32, revised manuscript), changed “To solve this problem, investigators have introduced light harvesting materials (such as dye, carbon materials and metal etc.) in constructing energy-storage system for photo-thermal conversion, and succeeded in converting light into heat, thus achieving efficient energy storage [10-14].”  (Line34, manuscript) to “To solve this problem, investigators have introduced light harvesting materials (such as dye, carbon materials and metal etc.) into phase change material systems to obtain high photo-thermal conversion and energy storage efficiency.” etc. (Line35, revised manuscript)

Reviewer 2 Report

Overall, the authors have done a good job of improving the manuscript. I feel that the enhanced introduction in particular makes a big difference. I feel that the manuscript can now be accepted for publication (but the English still needs to be significantly improved).

Author Response

QuestionBut the English still needs to be significantly improved.

Thank you for your suggestion.

Answer: We have asked native English speakers to help us revise our manuscript. And we have made many modifications. The modification of language is marked by “  ”

There are some substitutions of words, such as changing “Certified” (Line 82, manuscript) to “confirmed” (Line 79, revised manuscript), “Uninterrupted” (Line 15, manuscript) to “continuous” (Line 15, revised manuscript)  “Superiority” (Line 48, manuscript)  to “superiority performance” (Line 48, revised manuscript) etc.

There are some adjustments of sentences, for example, we changed “Therein thermal utilization exhibits inefficient direct heat usage factor owing to low thermal efficiency and inefficient utilization of visible light, which accounts for 44% of solar radiation.” (Line 32, manuscript) to “However, the thermal utilization suffers from inferior direct heat usage factor due to inefficient utilization of visible light, which accounts for 44% of solar radiation [8, 9].” (Line 32, revised manuscript), changed “To solve this problem, investigators have introduced light harvesting materials (such as dye, carbon materials and metal etc.) in constructing energy-storage system for photo-thermal conversion, and succeeded in converting light into heat, thus achieving efficient energy storage [10-14].”  (Line34, manuscript) to “To solve this problem, investigators have introduced light harvesting materials (such as dye, carbon materials and metal etc.) into phase change material systems to obtain high photo-thermal conversion and energy storage efficiency.” etc. (Line35, revised manuscript)

Reviewer 3 Report

Accepted with the performed changes, as the manuscript has improved a lot

Author Response

Thank you for your comments.